# HybridGBN-SR: A Deep 3D/2D Genome Graph-Based Network for Hyperspectral Image Classification

Haron C. Tinega [1], Enqing Chen [1,*], Long Ma [1], Divinah O. Nyasaka [2] and Richard M. Mariita [3]

1 School of Information Engineering, Zhengzhou University, Zhengzhou 450001, China; tinegaharon@gmail.com (H.C.T.); ielongma@zzu.edu.cn (L.M.)
2 The Kenya Forest Service, Nairobi P.O. Box 30513-00100, Kenya; dondieki@kenyaforestservice.org
3 Microbial BioSolutions, Troy, New York, NY 12180, USA; richard.mariita@microbialbiosolutions.com
* Correspondence: ieeqchen@zzu.edu.cn; Tel.: +86-371-6778-1544

**Abstract:** The successful application of deep learning approaches in remote sensing image classification requires large hyperspectral image (HSI) datasets to learn discriminative spectral–spatial features simultaneously. To date, the HSI datasets available for image classification are relatively small to train deep learning methods. This study proposes a deep 3D/2D genome graph-based network (abbreviated as HybridGBN-SR) that is computationally efficient and not prone to overfitting even with extremely few training sample data. At the feature extraction level, the HybridGBN-SR utilizes the three-dimensional (3D) and two-dimensional (2D) Genoblocks trained using very few samples while improving HSI classification accuracy. The design of a Genoblock is based on a biological genome graph. From the experimental results, the study shows that our model achieves better classification accuracy than the compared state-of-the-art methods over the three publicly available HSI benchmarking datasets such as the Indian Pines (IP), the University of Pavia (UP), and the Salinas Scene (SA). For instance, using only 5% labeled data for training in IP, and 1% in UP and SA, the overall classification accuracy of the proposed HybridGBN-SR is 97.42%, 97.85%, and 99.34%, respectively, which is better than the compared state-of-the-art methods.

**Keywords:** convolutional neural networks; genome graph; hyperspectral image classification; remote sensing; remote sensing image classification; residual learning; spectral–spatial features

## 1. Introduction

Remote sensing works by moving a vision system (satellite or aircraft) across the Earth's surface at various spatial resolutions and in different spectral bands of the magnetic spectrum to capture hyperspectral images (HSI) [1]. The vision system uses both imaging and spectroscopic methods to spatially locate specific components within the image scene under investigation based on their spectral features. The collected HSI data are a three-dimensional data structure with the x and y axes capturing the dimensions of the spatial images, and the z-axis is the number of spectral bands. Consequently, each pixel located on the x–y spatial domain contains a label representing the physical land cover of the target location [2].

For feature extraction and classification purposes, the voluminous spectral–spatial cues present in the HSI image represent an advantage in the detailed representation of the analyzed samples. However, they contain high spectral redundancy caused by significant interclass similarity and intraclass variability caused by changes in atmospheric, illumination, temporal, and environmental conditions, leading to data handling, storage, and analysis challenges [2]. For instance, an HSI system with a spatial resolution of $145 \times 145$ pixels will produce an image with 21,025 pixels for one spectral band. If the data contain 200 spectral bands, then a single image would produce over 4 million ($145 \times 145 \times 200$) data points. To overcome the challenges of spectral redundancy, most of

the HSI classification methods first employ dimensionality reduction methods to solve the curse of dimensionality introduced by spectral bands before extracting discriminative features from the resultant HSI data cube [3,4]. Some of the dimensionality reduction methods employed include, but are not limited to, the independent component analysis (ICA) [5], linear discriminant analysis (LDA) [6], and principal component analysis (PCA) [7]. Of all the aforementioned methods, PCA has become a popular dimensionality method among hyperspectral imaging [7–12]. Therefore, this paper uses PCA in dimensionality reduction.

This paper uses a convolutional neural network (CNN) in feature learning and extraction. Over the years, CNN has replaced rule-based methods because of its ability to extract reliable and effective features. CNN aims to extract highly discriminative features from input data [1]. Early feature extraction and learning experiments separately extracted spectral and spatial features, resulting in unsatisfactory classification results. Recent studies have recorded improved HSI classification accuracy when spectral–spatial features are simultaneously extracted, causing a shift of focus to developing models that utilize 3D convolutions in their network structure. Several researchers, such as Chen et al. [3] and Li et al. [8], simultaneously processed spectral–spatial features using the 3D-CNN model, which takes cubes of spatial size $7 \times 7$ and $5 \times 5$, respectively. Since then, numerous authors have implemented the 3D-CNN method to purposely extract deep spectral–spatial information concurrently [9,10]. Although the joint extraction of spectral–spatial features using 3D-CNNs achieved better classification accuracy, they are computationally expensive to be uniquely employed in HSI analysis and decrease in precision as the network deepens [11,12]. Several approaches have been proposed to address the challenge introduced by the 3D-CNNs to develop deep lightweight models that simultaneously process spectral–spatial cues for HSI classification. For instance, Roy et al. [13] replaced some 3D-CNN layers with the low-cost 2D-CNN in the network structure to develop a hybrid model that achieved state-of-the-art accuracies across all the HSI experimental datasets. Garifulla et al. [14] replaced the fully-connected (FC) layer with the global average pooling to reduce the network parameters and improve its inference speed [15]. Other researchers have used atrous (dilated) or deep-wise separable convolution instead of the conventional convolution in their network design to create lightweight models [16,17].

This paper extends the work of designing deep HSI classification models by proposing an optimal HybridGBN model variant abbreviated as HybridGBN-SR that trains on very few labeled training samples while increasing the classification accuracy. Unlike other methods that focus only on accuracy or speed, our network emphasizes the trade-off between these two; seemingly in a contrary aspect, the proposed model reduces the computation time while guaranteeing high classification accuracy. Therefore, the contributions of this paper are as follows: First, the proposed HybridGBN model variants utilize Genoblocks (a concept borrowed from biological genome graphs) in their network design. The Genoblocks contain identical and non-identical residual connections to enhance the feature learning of the HSI model even with very few training samples. Secondly, we further demonstrate the potential of residual learning in discriminative feature extraction by reinforcing the Genoblocks with various residual connectors, which resulted in the development of HybridGBN variants: HybridGBN-Vanilla, HybridGBN-SR, and HybridGBN-SSR. Lastly, we further the research of developing 3D/2D hybrid models in remote image classification to reduce model complexity.

The rest of this paper is organized as follows: Section 2 describes the proposed network; Section 3 contains the experimental setup and results discussion; Section 4 contains the conclusion of this research.

## 2. The Context of the Proposed Model

The proposed model seeks to extend research by developing deep models for HSI classification that can train on extremely few training samples while achieving high classification accuracy. The framework of the proposed model consists of preprocessing and feature learning classification steps as shown in Figure 1.

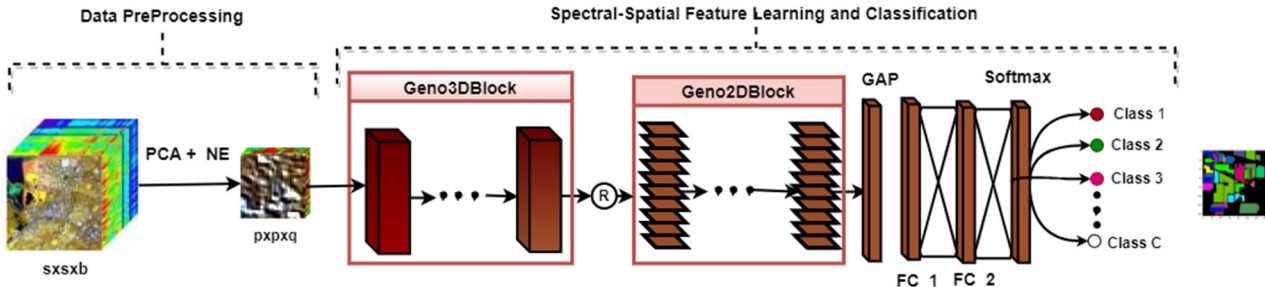

**Figure 1.** The framework of the proposed HybridGBN Model Variants.

In the data preprocessing step, as illustrated in Figure 1, the dimensionality of the original HSI data cube is reduced using the PCA method, and overlapping 3D patches are extracted using the neighborhood extraction approach. The extracted patches are then fed into the feature learning and classification step. First, the (Bottom) Geno3Dblock performs 3D convolutions on the input data. Then, its output is reshaped and once again fed into the second (top) Geno2Dblock, which performs 2D convolutions to extract more discriminative features. This approach was inspired by Roy et al. [13] in developing the HybridSN, which implements a bottom-heavy approach where 3D-CNN is employed at the bottom, followed by a spatial 2D-CNN at the top. Roy argues that the 3D-CNN at the bottom of the architecture facilitates the joint spectral–spatial feature representation, while the 2D-CNN on the top layers learns more abstract-level spatial representation. We then vectorize the feature maps of the last layer using global average pooling (GAP) [14] before forwarding to the FC layers and then to softmax layers for classification.

### 2.1. HSI Data Preprocessing Step

Assume a raw HSI data cube H with spatial dimensionality $k \in R^{s \times s}$ and b number of spectral bands. HSI data cube H can be viewed as a two-dimensional matrix $k \times b$ with each pixel composed of b spectral bands to form a one-hot label vector $V = (v1, v2, \ldots vz) \in R^{1 \times 1 \times c}$, where c denotes the class categories for each dataset. We apply the PCA method to reduce the data redundancy along the spectral dimension b in original HSI data cube H. The resulting HSI data cube I contains lesser spectral bands w such that $w < b$ while maintaining the spatial dimension k. We begin the PCA process by computing the covariance matrix, the product of the preprocessed data matrix, and its transpose (See line 2 of pseudo Algorithm 1). These steps aim to determine the variance of the input data variables from the mean with respect to each other to discern their correlation [18].

The following process involves the extraction of eigenvectors and eigenvalues associated with the covariance matrix to identify the principal components (See line 3). For each eigenvector, there is an eigenvalue, which indicates the variance in each principal component. The number of eigenvectors is equal to the number of eigenvalues, equivalent to the number of spectral bands b in the raw HSI data cube. Here, dimensionality reduction is attributed to the non-zero eigenvalues of the data matrix H of dimensionality $k \times b$.

The data matrix $H(k \times b)$ is decomposed using singular value decomposition (SVD) into $H = DEF^T$ where $D(k \times k)$ is the matrix of eigenvectors of the covariance matrix $HH^T$, $E(k \times b)$ is a diagonal matrix with eigenvalues as the main diagonal entries, and $F(b \times b)$ is the matrix of eigenvectors of the covariance matrix $H^TH$. Therefore, the total size for decomposition representation of H is $k \times k + k \times b + b \times b$, which is larger than $k \times b$, the size of H. Organizing information in principal components enables dimensionality reduction of spectral bands without losing valuable information. Therefore, the goal of PCA is to find an integer w smaller than b and use the first w columns of D while restricting E to the first w eigenvalues to show the effect of dimensionality reduction (See line 6).

| | **Algorithm 1: Spectral Data Reduction and Neighborhood Extraction** |
|---|---|
| 1 | Input: H(k × b) HSI data matrix, k pixels, b number of bands. |
| 2 | Compute the covariance matrix Q = $\frac{1}{B}$H$^T$H. |
| 3 | Compute the eigenvalues and eigenvector of *Q*. |
| 4 | Sort the eigenvectors to decrease eigenvalues: D, E, F, and normalize the columns to unity. |
| 5 | Make the diagonal entries of D and F non-negative. |
| 6 | Choose the value ww such that w < b. |
| 7 | Construct the transform matrix I(k × w) from the selected w eigenvectors. |
| 8 | Transform H(k × b) to I(k × w) in eigenspace to express data in terms of eigenvectors reduced from b to w. This gives a new set of basis vectors and a reduced b dimensional subspace of b vectors where data resides. |
| 9 | Reduced HSI data cube I will have dimensionality s × s × w, where w < b. |
| 10 | Perform neighborhood extraction on the new data cube I ∈ R$^{s×s×w}$. |
| 11 | Output: G number of small overlapping 3D patches of spatial dimension p × p and depth q. |

The computed eigenvectors are ranked in the descending order, i.e., from highest to the lowest in order of their eigenvalues, to find the principal components in order of significance. If we choose to keep w components (eigenvectors) out of b and discard the rest, we have a data matrix I(k × w), which can form a feature vector. Therefore, a feature vector is a matrix of vectors with eigenvectors of the components that we retain as columns. In this way, we have reduced the spectral dimensionality from b to w to form matrix I of k × w dimensions. Finally, we use this k × w eigenvector matrix to transform the samples to the new subspace. Applying PCA as a data reductionist method (see Figure 2) reduces dimensionality in the new space, not the original space [11].

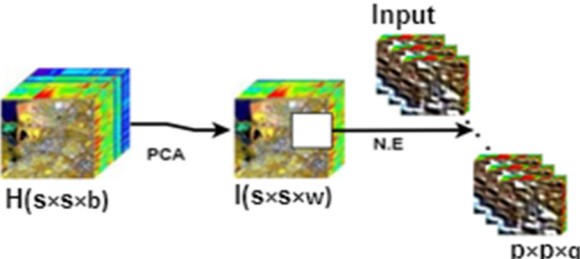

**Figure 2.** Preprocessing of raw HSI data cube.

The new data cube I ∈ R$^{s×s×w}$ is divided into G small overlapping 3D patches of spatial dimension p × p and depth q as shown in Figure 2. The label of the central pixel decides the truth labels at the spatial location (x, y).

### 2.2. Feature Extraction and Classification Step

This is the second step in our model design, as shown in Figure 1. We propose using a biological genome graph in feature extraction and classification and replacing 3D convolutions at the top of the network with low-cost 2D convolutions.

According to Manolov et al. [19], a tetraploid genome shown as variegated blocks (see Figure 3a) can be intertwined to form a complex pattern of the assembly graph without repeats or sequencing error (see Figure 3b). Graph genomics use graph-based alignment, which can correctly position all reads on the genome, as opposed to linear alignment, which is reference-based and cannot align all reads or use all of the available genome data. A graph genome is constructed from a population of genome sequences, such that a sequence path represents each haploid genome in this population through the graph [20]. Schatz et al. [21] and Rakocevic et al. [22] experimentally demonstrated that a graph genome could improve the volume of aligned reads, resolve haplotypes, and create a more accurate depiction of population diversity [18,20]. In this perspective, we propose HybridGBN models that utilize Genoblocks, a concept borrowed from genomics, in their network design.

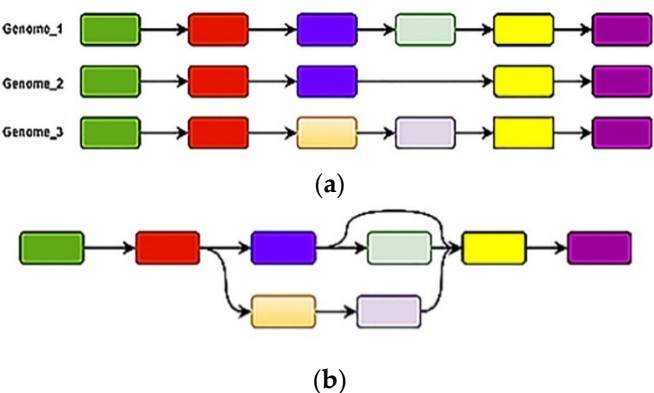

**(a)**

**(b)**

**Figure 3.** (**a**) tetraploid genome (**b**) assembly graph.

### 2.2.1. The Architecture of Genoblocks

The Genoblocks use CNNs in their design. The CNNs have three parts: the input, hidden, and output layers. The role of the hidden (convolutional) layer is to perform the convolution operation, i.e., transforming the input received into some form and passing it to the next layer without losing its characteristics.

Mathematically, an individual neuron is computed by striding a weight filter T with bias n on a vector of inputs E to produce an output feature map m. The term striding in CNN refers to the number of pixels (in integer) by which the filter window shifts (either from left to right and from top to bottom) after each operation until all pixels are convolved. Mathematically, this can be explained as follows

$$m = f(TE + n) \tag{1}$$

where $f(°)$ is a nonlinear function used as an activation function to introduce the nonlinearity.

We use the rectified linear unit (ReLU) function since it is more efficient than the sigmoid function in the convergence of the training procedure [23]. The ReLU function is defined as follows

$$f = \max(0, x) \tag{2}$$

Research in computer vision has shown that the depth of the network has a higher advantage than the width of the network in terms of better feature learning and fitting [15,24]. Successful training of deep networks using small samples can be realized through residual connections [11]. Works by Mou et al. [24] and Zhong et al. [10] exhibited extensive network residual learning (RL) models to extract additional discriminative characteristics for HSI classification [11] to sufficiently solve the degradation problem profound in deep networks. Hence, the strength of Genoblocks lies in its utilization of the residual connections and multi-scale kernels that extract abundant contextual features to attain a high rate of generalizability [25]. A vanilla Genoblock shown in Figure 4 utilizes identical and non-identical residual connections to recover lost features during convolution

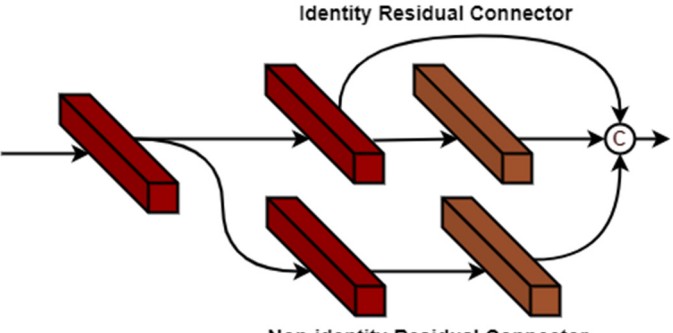

**Figure 4.** Identity and non-identity residual network in a vanilla Genoblock.

### 2.2.2. The Genoblock Variants

The Geno3Dblock is the first (bottom) block in the structure of the proposed HybridGBN model. Figure 4 illustrates the basic Genoblock from which we created the three variants: Geno3Dblock-Vanilla (see Figure 5), Geno3Dblock-SR (see Figure 6), and Geno3Dblock-SSR (see Figure 7). We use ReLU as the activation function for each convolution layer. Therefore, the activation value of these Geno3Dblock variants at spectral–spatial position $(x, y, z)$ in the $j^{th}$ feature map of the $i^{th}$ layer is denoted as $v_{i,j}^{x,y,z}$, and is given by

$$v_{i,j}^{x,y,z} = f(n_{i,j} + (T \otimes E)_{i,j}) \tag{3}$$

where parameter $n_{i,j}$ is the bias value for the $j^{th}$ feature map of the $i^{th}$ layer, T is the kernel function with the learned weights, E is the input or the layer, and $\otimes$ denotes the convolution operator.

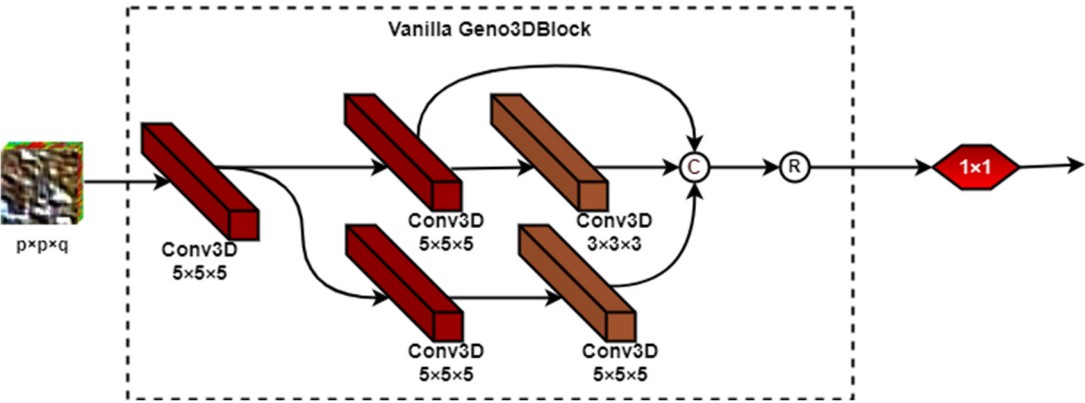

**Figure 5.** Framework of Geno3Dblock-Vanilla.

Similarly, the convolution operator $(T \otimes E)_{i,j}$ is given by

$$(T \otimes E)_{i,j} = \sum_{m=1}^{M} \sum_{r=0}^{R_i-1} \sum_{q=0}^{Q_i-1} \sum_{p=0}^{P-1} w_{i,j,m}^{r,q,p} \times v_{(i-1),m}^{(x+r),(y+q),(z+p)} \tag{4}$$

Parameters $R_i$, $Q_i$, and $P_i$ denote the kernel width, height, and depth dimensions, respectively. M is the total number of feature maps in the $(i-1)^{th}$ layer connected to the current feature map. $w_{i,j,m}^{r,q,p}$ is the value of the weight parameter for position $(r, q, p)$ kernel connected to the $m^{th}$ feature map in the previous layer.

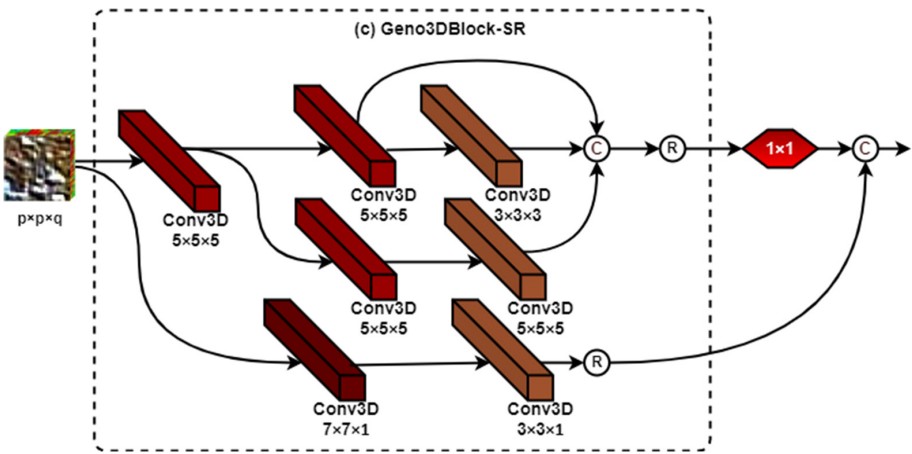

**Figure 6.** Framework of Geno3Dblock-SR.

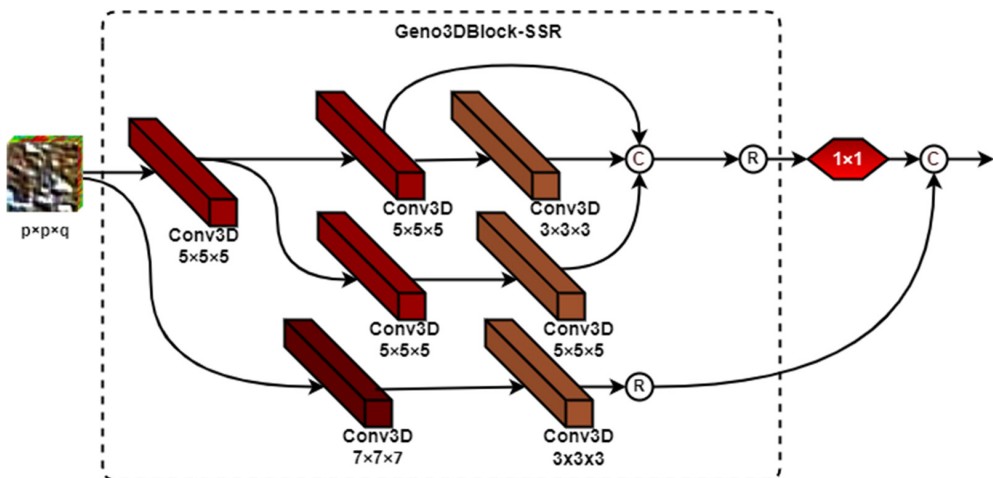

**Figure 7.** Framework of Geno3Dblock-SSR.

We apply padding P to facilitate the use of an identical residue connection that requires preserving the input image dimensions. In CNN, padding refers to the number of pixels added to the border of an image when the kernel processes it to avoid shrinking. Padding is vital in image processing using CNN as it extends the image area, which assists the kernel in producing more accurate image analyses. We can perform padding by either replicating the edge of the original image or zero padding. Zero padding is a popular technique to pad the input volume with zeros. Zero padding an output image O for any given layer is given by

$$O = [(G - F + 2P)S] \times [(H - F + 2P)S] \times [D_y] \tag{5}$$

where O is the output dimension, G is the width of the input, H is the height of the input, F is the filter size, P is the padding, and S is the stride. $D_y$ is the depth of the output image.

The Geno3Dblock-Vanilla: This block utilizes multi-scale kernels (i.e., $3 \times 3 \times 3$, $5 \times 5 \times 5$) to extract multi-scale features from the image map. The structure of the Geno3Dblock-Vanilla is as shown in Figure 5. This is the basic building block used to develop the HybridGBN-Vanilla model.

The Geno3Dblock-SR: This block adds an extra spatial residual (SR) connection to the basic building block (Geno3Dblock-Vanilla), as shown in Figure 6. This block is used in the development of the HybridGBN-SR model.

The Geno3Dblock-SSR: Here, the spatial residual (SR) connection in Geno3Dblock-SR is replaced with a spectral–spatial residual (SSR) connector, as shown in Figure 7. We utilized this block in the development of the HybridGBN-SSR model.

The output of the above Geno3Dblocks is reshaped before passing to the top Geno2Dblock for further feature learning. Reshaping is the deformation of 3D features to 2D features to reduce the model operational cost. For instance, a convolutional layer with 64 feature map data of the size of $3 \times 3 \times 3$ can be reshaped into 192 2D feature maps of size $3 \times 3$, as shown in Figure 8.

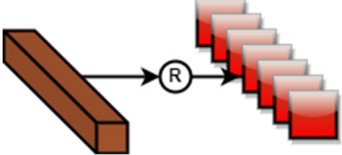

**Figure 8.** Framework for 3D to 2D feature deformation.

Geno2Dblock: To reduce the model complexity, we developed the Geno2Dblock shown in Figure 9, which is used in the second (top) block of the proposed HybridGBN model variants to learn more discriminative spatial features. It utilizes maxpooling2D and

dilated convolution arranged in parallel to capture the context information and multi-scale features [26].

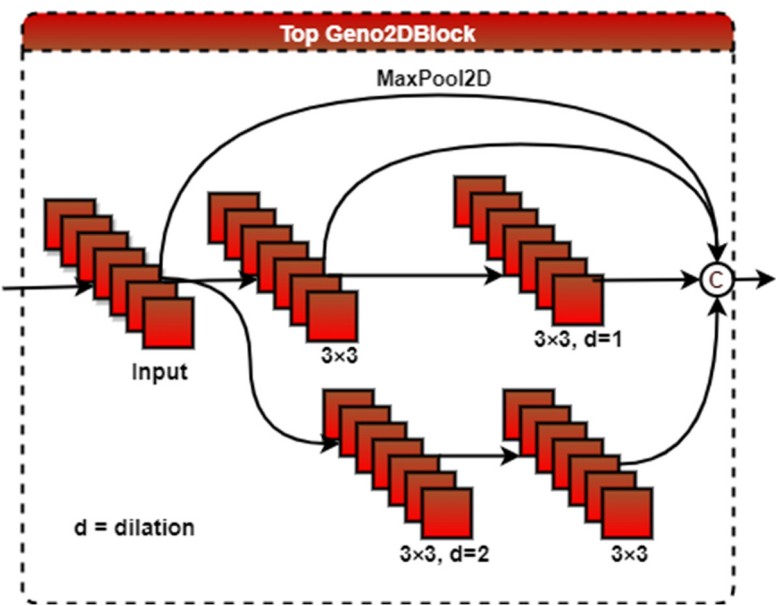

**Figure 9.** The Framework of Geno2Dblock.

The activation value of the 2D convolution in the Geno2Dblock at the $i^{th}$ layer at spatial position $(x, y)$ in the $j^{th}$ feature map is given by

$$v_{i,j}^{x,y} = f\left(n_{i,j} + (T \otimes E)_{i,j}\right) \qquad (6)$$

The convolution operator $(T \otimes E)_{i,j}$ for a 2D layer at spatial position $(x, y)$ in the $j^{th}$ feature map can be further explained as

$$(T \otimes E)_{i,j} = \sum_{m=1}^{M} \sum_{r=0}^{R_i-1} \sum_{q=0}^{Q_i-1} w_{i,j,m}^{r,q} \times v_{(i-1),m}^{(x+r),(y+q)} \qquad (7)$$

where, $w_{i,j,m}^{r,q}$ is the weight for spatial position $(r, q)$ kernel connected to the previous layer's $m^{th}$ feature map.

We utilized max2Dpooling in the Geno2Dblock to help recover lost features during the convolution process and control overfitting. The max2Dpooling function partitions the input feature map into a set of rectangles and outputs the maximum value for each sub-region. Mathematically, the general pooling function can be computed as follows

$$cZ_l^k = g_p\left(cF_l^k\right) \qquad (8)$$

where $cZ_l^k$ represents the pooled feature map of $l^{th}$ layer for $k^{th}$ input feature map $cF_l^k$, and $g_p(.)$ defines the type of pooling operation. In this research, $g_p(.)$ is a MaxPooling2D.

In place of flattening, we used the global average pooling (GAP) to reduce the number of network parameters and effectively avert the model from overfitting. GAP achieves this by reducing each h × w feature map to a single number by taking the average of all hw values. We then used two FC layers to learn more discriminative features further.

The output from the FC layers is then passed to the softmax layer to perform classification. The softmax function is a probabilistic-based function that uses a probability score

to measure the correlation between output and reference values. Therefore, the probability that a given input belongs to the class c label of the HSI dataset is given by

$$p(y_i) = \frac{e^{y_i}}{\sum_{j=1}^{C} e^{y_j}} p(y_i)$$

$$\text{for } i = 1, \ldots c, \ldots, C, \text{ and } y = y_1, \ldots, y_C \in R^C \tag{9}$$

where $y_i = 1, \ldots$, C are the target ground truth values of the input vector to the softmax function and $p(y_i)$ as the output class membership distribution with i as the index of the test pixel.

The number of kernels of the last layer is set to equal the number of classes defined in the HSI dataset under study.

We can treat the whole procedure of training HybridGBN model variants as optimizing parameters to minimize the multiclass loss function between the network outputs and the ground truth values for the training data set. The network is fine-tuned through the backpropagation. The loss function L is given by

$$L = \sum_{i=1}^{N} \sum_{c=1}^{C} \max(0, 1 - p(y_i)) \tag{10}$$

Finally, the prediction label is decided by taking the *argmin* value $\hat{y}_i$ of the loss function

$$\hat{y}_i = \underset{c}{\text{argmin } L} \tag{11}$$

## 3. Experimental Results and Discussion

In this section, we report the quantitative and qualitative results of the proposed HybridGBN model variants in comparison with the other state-of-the-art methods such as 2D-CNN, M3D-DCN, HybridSN, R-HybridSN, and SSRN over the selected publicly available HSI datasets, namely Indian Pines (IP), University of Pavia (UP), and Salinas Scene (SA). This section is divided into Sections 3.1–3.6 that describe experimental datasets, experimental setup, evaluation criteria, experimental results, and discussions on very small training sample data, varying training sample data, and the time complexity of the selected models over IP, UP, and SA datasets.

### 3.1. Experimental Datasets

The IP dataset was collected by the airborne visible/infrared imaging spectrometer (AVIRIS) sensor flying over the IP test site in Northwestern Indiana. The original image is $145 \times 145 \times 220$ in dimension. After discarding 20 spectral bands due to water absorption, the data used in this experiment have the size of $145 \times 145 \times 200$. The ground truth of the IP scene dataset consists of 16 not mutually exclusive labels [27].

The UP dataset was collected by a Reflective Optics System Imaging Spectrometer (ROSIS) flying over Pavia city, northern Italy. The spectral–spatial dimension of the original HSI image is $610 \times 340 \times 1155$. We reduced these dimensions to $610 \times 340 \times 103$ by eliminating 12 noisy bands [1,18]. The UP dataset has nine classes; except for one class (Shadows), the rest of the classes have more than 1000 labeled pixels.

The SA dataset was captured by the AVIRIS sensor flying over the Salinas Valley, California. The original size of the HSI image is $512 \times 217 \times 224$. After eliminating 20 bands covering the water absorption region, the resultant image size used in this experiment is $512 \times 217 \times 204$. The land cover has been categorized into 16 class labels [27].

### 3.2. Experimental Setup

All experiments were conducted online using Google Colab Inc. We split our datasets into training and testing sets. We report the results as the average of seven runs. Moreover, we applied the grid search method to select the best optimizer method, learning rate, dropout,

and epochs for the proposed method. For all the datasets, we chose Adam optimizer with learning rates of 0.0005, 0.0007, and 0.001 for IP, UP, and SA datasets, respectively. The optimal dropout for IP, UP, and SA was 0.35, 0.5, and 0.4, respectively, while the optimal epochs for IP, UP, and SA were 100, 150, and 100. Using HybridGBN-Vanilla as the basic building block, we varied the spatial window size over IP, UP, and SA datasets to obtain the optimal window size. Considering the overall accuracy (OA), average accuracy (AA), and Kappa coefficient (Kappa), the optimal spatial window size of the HybridGBN-Vanilla over IP, UP, and SA datasets is $19 \times 19$, $15 \times 15$, and $23 \times 23$ respectively. Therefore, the dimensions of the overlapping 3D patches of the input volume are set to $19 \times 19 \times 30$, $15 \times 15 \times 15$, and $23 \times 23 \times 15$, respectively. We used the same window size on HybridGBN variants (e.g., HybridGBN-SSR and HybridGBN-SR) for a fair comparison.

*3.3. Evaluation Criteria*

To assess the performance of the proposed HSI models, we use the Kappa, OA, and AA evaluation measures.

The OA represents the percentage of correctly classified samples, with 100% accuracy being a perfect classification where all samples were classified correctly. It is given by

$$OA = \frac{\text{Correctly classified samples}}{\text{The total number of samples}} \tag{12}$$

where Correctly classified samples are cases where the predicted results are the same as the actual ground truth label.

The AA gives the mean result of per class classification accuracies, and it is given by

$$AA = \frac{1}{c} \sum_{i=1}^{C} (x) \tag{13}$$

where c is the number of classes, and x is the percentage of correctly classified pixels in a single class.

The Kappa provides information on what percentage of the classification map concurs with the ground truth map, and it is given by

$$Kappa = \frac{P_o - P_e}{1 - P_e} \tag{14}$$

$P_o$ denotes the observed agreement, which is the model classification accuracy, and $P_e$ symbolizes the expected agreement between the model classification map and the ground truth map by chance probability. When the Kappa value is 1, it indicates perfect agreement, while 0 indicates agreement by chance.

*3.4. Experimental Results and Discussions on Very Small Training Sample Data*

This section aims to show the robustness of the models on very little training sample data, i.e., 5% for IP, and 1% for UP and SA datasets, respectively. We use the remaining sample data portion for testing.

3.4.1. Distribution of the Training and Testing Sample Data over IP, UP, and SA Datasets on Very Little Sample Data

Tables 1–3 provide the detailed distribution of the training and testing samples of IP, UP, and SA datasets.

**Table 1.** Per Class information for IP dataset.

| Class No | Class Label | Total Samples (Pixels) | Total Samples (%) | Training | Testing |
|---|---|---|---|---|---|
| 1 | Alfalfa | 46 | 0.45 | 2 | 44 |
| 2 | Corn-notill | 1428 | 13.93 | 71 | 1357 |
| 3 | Corn-mintill | 830 | 8.1 | 41 | 789 |
| 4 | Corn | 237 | 2.31 | 12 | 225 |
| 5 | Grass-pasture | 483 | 4.71 | 24 | 459 |
| 6 | Grass-trees | 730 | 7.12 | 37 | 693 |
| 7 | Grass-pasture-mowed | 28 | 0.27 | 1 | 27 |
| 8 | Hay-windrowed | 478 | 4.66 | 24 | 454 |
| 9 | Oats | 20 | 0.2 | 1 | 19 |
| 10 | Soybean-notill | 972 | 9.48 | 49 | 923 |
| 11 | Soybean-mintill | 2455 | 23.95 | 123 | 2332 |
| 12 | Soybean-clean | 593 | 5.79 | 30 | 563 |
| 13 | Wheat | 205 | 2 | 10 | 195 |
| 14 | Woods | 1265 | 12.34 | 63 | 1202 |
| 15 | Buildings-Grass-Trees-Drives | 386 | 3.77 | 19 | 367 |
| 16 | Stone-Steel-Towers | 93 | 0.91 | 5 | 88 |

**Table 2.** Per Class information for the UP dataset.

| Class No | Class Label | Total Samples (Pixels) | Total Samples (%) | Training | Testing |
|---|---|---|---|---|---|
| 1 | Asphalt | 6631 | 15.5 | 66 | 6565 |
| 2 | Meadows | 18,649 | 43.6 | 186 | 18,463 |
| 3 | Gravel | 2099 | 4.91 | 21 | 2078 |
| 4 | Trees | 3064 | 7.16 | 31 | 3033 |
| 5 | Painted | 1345 | 3.14 | 13 | 1332 |
| 6 | Bare | 5029 | 11.76 | 50 | 4979 |
| 7 | Bitumen | 1330 | 3.11 | 13 | 1317 |
| 8 | Self-Blocking | 3682 | 8.61 | 37 | 3645 |
| 9 | Shadows | 947 | 2.21 | 10 | 937 |

**Table 3.** Per Class information for the SA dataset.

| Class No | Class Label | Total Samples (Pixels) | Total Samples (%) | Training | Testing |
|---|---|---|---|---|---|
| 1 | Broccoli_green_weeds_1 | 2009 | 3.71 | 20 | 1989 |
| 2 | Broccoli_green_weeds_2 | 3726 | 6.88 | 37 | 3689 |
| 3 | Fallow | 1976 | 3.65 | 20 | 1956 |
| 4 | Fallow_rough_plow | 1394 | 2.58 | 14 | 1380 |
| 5 | Fallow_smooth | 2678 | 4.95 | 27 | 2651 |
| 6 | Stubble | 3959 | 7.31 | 39 | 3920 |
| 7 | Celery | 3579 | 6.61 | 36 | 3543 |
| 8 | Grapes_untrained | 11,271 | 20.82 | 113 | 11,158 |
| 9 | Soil_vineyard_develop | 6203 | 11.46 | 62 | 6141 |
| 10 | Corn_senesced_green_weeds | 3278 | 6.06 | 33 | 3245 |
| 11 | Lettuce_romaine_4wk | 1068 | 1.97 | 11 | 1057 |
| 12 | Lettuce_romaine_5wk | 1927 | 3.56 | 19 | 1908 |
| 13 | Lettuce_romaine_6wk | 916 | 1.69 | 9 | 907 |
| 14 | Lettuce_romaine_7wk | 1070 | 1.98 | 11 | 1059 |
| 15 | Vineyard_untrained | 7268 | 13.43 | 72 | 7196 |
| 16 | Vineyard_vertical_trellis | 1807 | 3.34 | 18 | 1789 |

From Table 1, we can observe that the IP dataset is unbalanced, with some classes having one or two training samples when a minimal sample size of 5% is chosen. Table 2 shows that the UP dataset is a slightly balanced dataset with most classes well represented even at minimal training sample data of 1%. Hence, we expect the classifiers to have better classification accuracies than the IP dataset. We can see in Table 3 that all classes are well represented at 1% training sample data for the SA dataset. Therefore, we conclude that the IP is the most unstable dataset, followed by the UP and SA datasets.

3.4.2. The Performance of Selected Models over IP, UP, and SA Datasets Using Very Limited Training Sample Data

This subsection presents per class accuracy, the Kappa, OA, and AA of the compared methods in an extreme condition of very small sample data over IP, UP, and SA datasets, as shown in Tables 4–6.

**Table 4.** The Kappa, OA, and AA results in a percentage of the compared models at 5% training sample data over IP dataset.

| Class | 2D-CNN | M3D-DCNN | HybridSN | R-HybridSN | SSRN | HybridGBN-Vanilla | HybridGBN-SSR | HybridGBN-SR |
|---|---|---|---|---|---|---|---|---|
| 1 | 7.95 | 27.5 | 61.82 | 45 | 12.99 | 84.68 | 81.13 | 83.12 |
| 2 | 70.69 | 59.15 | 92.25 | 95.45 | 93.04 | 94.76 | 95.96 | 95.55 |
| 3 | 52.84 | 45.07 | 92.97 | 97.36 | 93.72 | 98.89 | 98.58 | 99.51 |
| 4 | 27.51 | 38.49 | 78.22 | 94.8 | 72.38 | 93.44 | 93.7 | 96.38 |
| 5 | 90.44 | 70.33 | 96.6 | 98.85 | 98.16 | 99.69 | 99.72 | 99.6 |
| 6 | 98.59 | 97.2 | 98.11 | 99.32 | 99.86 | 99.07 | 98.99 | 99.09 |
| 7 | 10.37 | 18.52 | 68.52 | 95.56 | 0 | 98.37 | 95.77 | 99.47 |
| 8 | 99.96 | 98.04 | 99.96 | 100 | 99.94 | 100 | 100 | 100 |
| 9 | 16.32 | 25.79 | 83.68 | 65.26 | 0 | 64.66 | 76.69 | 78.2 |
| 10 | 67.84 | 55.85 | 96.12 | 95.9 | 91.01 | 97.83 | 97.29 | 96.61 |
| 11 | 78.16 | 76.2 | 96.66 | 98.09 | 95.63 | 97.89 | 98.36 | 98.21 |
| 12 | 42.01 | 33.89 | 85.44 | 89.15 | 87.9 | 90.57 | 91.29 | 92.46 |
| 13 | 98.97 | 91.23 | 94.97 | 99.74 | 98.53 | 97.05 | 98.53 | 98.1 |
| 14 | 97.65 | 94.68 | 99.34 | 99.26 | 99.82 | 99.56 | 99.3 | 99.73 |
| 15 | 62.62 | 42.37 | 82.92 | 87.66 | 82.09 | 94.36 | 92.76 | 92.41 |
| 16 | 76.02 | 49.32 | 80 | 88.18 | 82.31 | 91.52 | 91.06 | 89.94 |
| Kappa | 0.718 ± 0.01 | 0.642 ± 0.045 | 0.934 ± 0.012 | 0.96 ± 0.004 | 0.923 ± 0.49 | 0.968 ± 0.43 | 0.97 ± 0.4 | 0.971 ± 0.25 |
| OA (%) | 75.47 ± 0.81 | 68.88 ± 3.77 | 94.24 ± 1.01 | 96.46 ± 0.33 | 93.39 ± 0.43 | 97.15 ± 0.38 | 97.32 ± 0.35 | 97.42 ± 0.22 |
| AA (%) | 62.37 ± 1.64 | 57.73 ± 6.52 | 87.97 ± 1.93 | 90.6 ± 1.53 | 75.28 ± 1.25 | 93.9 ± 1.11 | 94.32 ± 1.89 | 94.9 ± 2.4 |

**Table 5.** The Kappa, OA, and AA results in a percentage of the compared models at 1% training sample data over the UP dataset.

| Class | 2D-CNN | M3D-DCNN | HybridSN | R-HybridSN | SSRN | HybridGBN-Vanilla | HybridGBN-SSR | HybridGBN-SR |
|---|---|---|---|---|---|---|---|---|
| 1 | 96.88 | 90.56 | 95.72 | 96.94 | 98.76 | 97.54 | 98.02 | 98.13 |
| 2 | 99.01 | 89.47 | 99.68 | 99.69 | 99.91 | 99.65 | 99.81 | 99.55 |
| 3 | 75.08 | 59.11 | 84.38 | 87.17 | 85.72 | 90.77 | 90.67 | 93.81 |
| 4 | 87.74 | 93.25 | 87.7 | 89.15 | 94.85 | 90.23 | 87.49 | 91.07 |
| 5 | 98.17 | 93.66 | 98.99 | 99.51 | 99.76 | 99.75 | 99.5 | 99.09 |
| 6 | 75.51 | 69.63 | 96.82 | 98.44 | 96.11 | 97.55 | 97.49 | 98.85 |
| 7 | 61.32 | 65.71 | 84.42 | 95.82 | 95.98 | 99.29 | 95.75 | 99.44 |
| 8 | 80.61 | 78.35 | 89.18 | 93.28 | 94.96 | 93.8 | 92.22 | 95.82 |
| 9 | 97.97 | 94.41 | 71.71 | 77.82 | 99.89 | 91.65 | 94.22 | 92.07 |
| Kappa | 0.881 ± 0.008 | 0.798 ± 0.016 | 0.935 ± 0.011 | 0.955 ± 0.007 | 0.97 ± 0.54 | 0.964 ± 0.56 | 0.960 ± 0.82 | 0.972 ± 0.53 |
| OA (%) | 91.13 ± 0.55 | 84.63 ± 1.21 | 95.09 ± 0.8 | 96.59 ± 0.5 | 97.67 ± 0.4 | 97.28 ± 0.42 | 97.02 ± 0.61 | 97.85 ± 0.4 |
| AA (%) | 85.81 ± 1.48 | 81.57 ± 1.79 | 89.84 ± 1.93 | 93.09 ± 1.2 | 96.22 ± 0.82 | 95.58 ± 0.79 | 95.02 ± 1.26 | 96.42 ± 0.54 |

We observe in Tables 4 and 6 that the Kappa, OA, and AA of the proposed HybridGBN variants (HybridGBN-Vanilla, HybridGBN-SR, and HybridGBN-SSR) are higher than the compared state-of-the-art methods such as the 2D-CNN, M3D-DCNN, SSRN, R-HybridSN, and HybridSN for IP and SA datasets. However, only the proposed HybridGBN-SR method records superior performance in the UP dataset over all the compared models.

Across all the datasets, the M3D-DCNN recorded the lowest classification accuracy because it mainly uses multi-scale 3D dense blocks in its network structure, which is prone to overfitting when subjected to limited training sample data. We note that the accuracy of 2D-CNN is slightly higher than that of M3D-DCNN due to its ability to extract more discriminative spatial features vital to HSI classification. On the other hand, HybridSN [13] utilizes both 3D-CNNs and 2D-CNN in the HSI network structure. The SSRN posits better classification accuracy than 2D-CNN and M3D-CNN in all datasets. It defeats HybridSN in the UP dataset because it uses skip connections to extract deep features, effectively

addressing the degradation problem in the deep depth network. The R-HybridSN [28] achieved better classification performance than all the earlier mentioned methods because it utilizes non-identical multi-scale residual connections in its network structure.

**Table 6.** The Kappa, OA, and AA results in a percentage of the compared models at 1% training sample data over the SA dataset.

| Class | 2D-CNN | M3D-DCNN | HybridSN | R-HybridSN | SSRN | HybridGBN-Vanilla | HybridGBN-SSR | HybridGBN-SR |
|---|---|---|---|---|---|---|---|---|
| 1 | 99.97 | 94.88 | 99.99 | 100 | 100 | 100 | 100 | 100 |
| 2 | 99.86 | 99.61 | 100 | 99.97 | 100 | 100 | 100 | 100 |
| 3 | 99.43 | 91.89 | 99.82 | 99.49 | 99.96 | 99.92 | 99.97 | 100 |
| 4 | 98.83 | 98.33 | 98.38 | 98.72 | 99.72 | 99.23 | 97.64 | 99.67 |
| 5 | 96.77 | 98.83 | 99.26 | 98.43 | 98.73 | 98.43 | 98.66 | 99 |
| 6 | 99.79 | 98.09 | 99.93 | 99.9 | 100 | 99.89 | 99.89 | 99.71 |
| 7 | 99.33 | 97.67 | 99.95 | 99.96 | 99.99 | 99.95 | 99.94 | 100 |
| 8 | 87.39 | 82.4 | 97.77 | 98.23 | 95.06 | 99.75 | 99.64 | 99.69 |
| 9 | 99.97 | 98.14 | 99.99 | 99.99 | 100 | 100 | 100 | 100 |
| 10 | 93.98 | 87.6 | 98.36 | 97.9 | 98.33 | 98.78 | 98.78 | 99 |
| 11 | 89.62 | 86.72 | 96.06 | 96.46 | 97.42 | 99.53 | 98.72 | 99.18 |
| 12 | 99.99 | 96.99 | 97.44 | 99.09 | 100 | 99.98 | 99.59 | 99.69 |
| 13 | 98.52 | 97.14 | 97.42 | 82.82 | 93.02 | 82.89 | 85.42 | 93.89 |
| 14 | 97.64 | 91.78 | 99.52 | 97.25 | 95.62 | 94.94 | 98.03 | 95.71 |
| 15 | 79.46 | 64.42 | 97.06 | 95.12 | 88.18 | 97.2 | 98.31 | 98.21 |
| 16 | 95.71 | 78.14 | 100 | 99.71 | 99.49 | 99.98 | 99.98 | 99.96 |
| Kappa | 0.928 ± 0.003 | 0.867 ± 0.002 | 0.985 ± 0.007 | 0.98 ± 0.004 | 0.966 ± 0.61 | 0.989 ± 0.61 | 0.991 ± 0.34 | 0.993 ± 0.16 |
| OA (%) | 93.55 ± 0.26 | 88.02 ± 1.35 | 98.72 ± 0.59 | 98.25 ± 0.4 | 96.94 ± 0.55 | 98.91 ± 0.55 | 99.16 ± 0.31 | 99.34 ± 0.14 |
| AA (%) | 96.02 ± 0.42 | 91.41 ± 0.81 | 98.81 ± 0.5 | 97.69 ± 0.69 | 97.84 ± 0.52 | 97.84 ± 1 | 98.41 ± 1.06 | 98.98 ± 0.32 |

Comparing the HybridGBN variants, we observe that the classification performance of HybridGBN-SR and HybridGBN-SSR is better than that of HybridGBN-Vanilla. Hence, adding a non-identical residual connection to the basic building block can significantly improve the classification accuracy across all the experimental datasets. For example, in Table 4, we can observe that the proposed HybridGBN-SR improves OA and AA of HybridGBN-SSR by +0.1% and +0.58%, respectively, and that of HybridGBN-Vanilla by +0.27% and +1%, respectively, in the IP dataset. In the UP dataset (see Table 5), the proposed HybridGBN-SR increases the OA and AA of HybridGBN-SSR by +0.83% and +1.4%, and that of HybridGBN-Vanilla by +0.57% and +0.84%, respectively. In SA (see Table 6), the OA and AA of the proposed HybridGBN-SR are higher than HybridGBN-SSR by +0.18% and +0.57%, and higher than HybridGBN-Vanilla by +0.43% and +1.14%, respectively. We can observe a significant improvement in average accuracy compared with the overall accuracy. The HybridGBN-SR performs better on classes with meager training sample data. For instance, at class 7 of the IP dataset, the HybridGBN-SR improves the per class accuracy of 2D-CNN, M3D-CNN, SSRN, HybridSN, and R-HybridSN by +89.1%, +80.95%, +99.47%, +30.95%, and +3.91%, respectively. The difference in performance between HybridGBN-SSR and HybridGBN-SR can be attributed to the additional residual connections at the bottom block of the network. In HybridGBN-SSR, the additional residual connection simultaneously extracts spatial–spectral features. Unlike in HybridGBN-SSR, the additional residual connection in HybridGBN-SR extracts spatial features only while preserving raw spectral features, resulting in the convolution of high-level spatial features with low-level spectral features in the top network layers. To the best of our knowledge, this leads to the extraction of more discriminative features and, hence, increases classification accuracy.

In comparison with other state-of-the-art methods the HybridGBN-SR improves the overall accuracy of 2D-CNN, M3D-CNN, SSRN, HybridSN, and R-HybridSN by +21.95, +28.54, +4.03, +3.18, +0.96 on the IP dataset (see Table 4), +6.72%, +3.22%, +2.76%, +1.26%, +0.18% on the UP dataset (see Table 5), and +5.79%, +11.32%, +0.62%, +1.09%,+2.4% on the SA dataset (see Table 6), respectively. This trend is more pronounced in classes with less than 1% training sample points. In this perspective, we propose the HybridGBN-SR

that can learn more discriminate features at extremely small training data samples and in unbalanced datasets.

3.4.3. Training Accuracy and Loss Graph of the Selected Models on Very Limited Sample Data

We can observe in Figures 10–12 that the proposed HybridGBN-SR converges better than SSRN and HybridSN and worse than R-HybridSN over the IP and UP datasets. The HybridGBN-SR training and loss graph in the SA dataset is comparable to R-HybridSN and HybridSN, indicating its competitiveness over the SA dataset (See Figure 12).

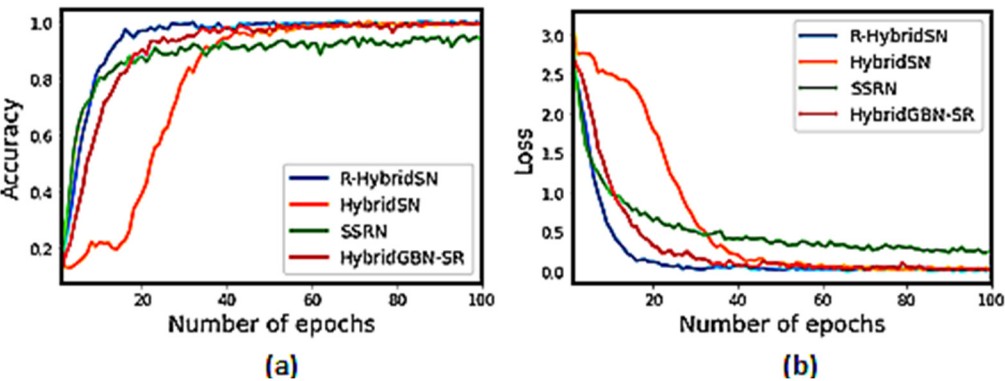

**Figure 10.** Training graphs for R-HybridSN, HybridSN, SSRN, and HybridGBN-SR for each epoch over IP dataset: (**a**) The training accuracy graph; (**b**) The loss convergence graph.

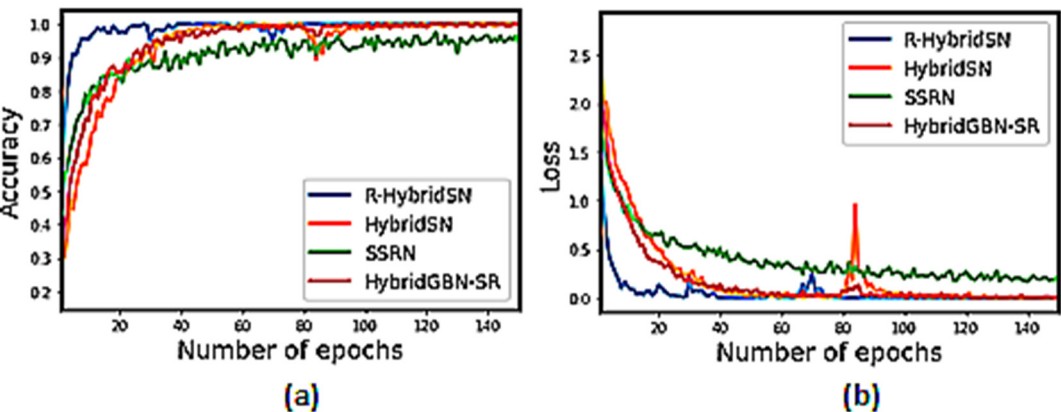

**Figure 11.** Training graphs for R-HybridSN, HybridSN, SSRN, and HybridGBN-SR for each epoch over UP dataset: (**a**) The training accuracy graph; (**b**) The loss convergence graph.

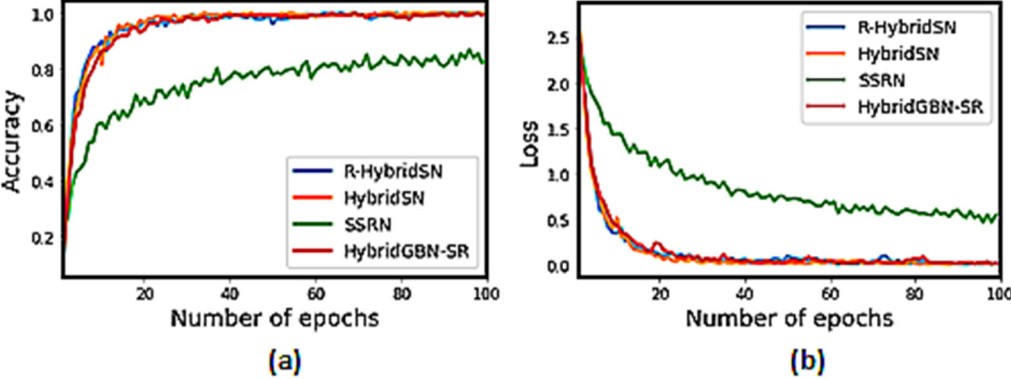

**Figure 12.** Training graphs for R-HybridSN, HybridSN, SSRN, and HybridGBN-SR for each epoch over SA dataset: (**a**) The training accuracy graph; (**b**) The loss convergence graph.

### 3.4.4. Confusion Matrix

This subsection further demonstrates the competitiveness of the proposed HybridGBN-SR using the confusion matrix over the IP, UP, and SA datasets.

With a closer look at the confusion matrix in Figures 13–15 we can observe that most of the sample data of the proposed HybridGBN-SR lie in the diagonal line even with limited training data compared to SSRN, HybridSN, and R-HybridSN over the IP, UP, and SA datasets. Therefore, the proposed model correctly classified most sample data, demonstrating its robustness over a small training data.

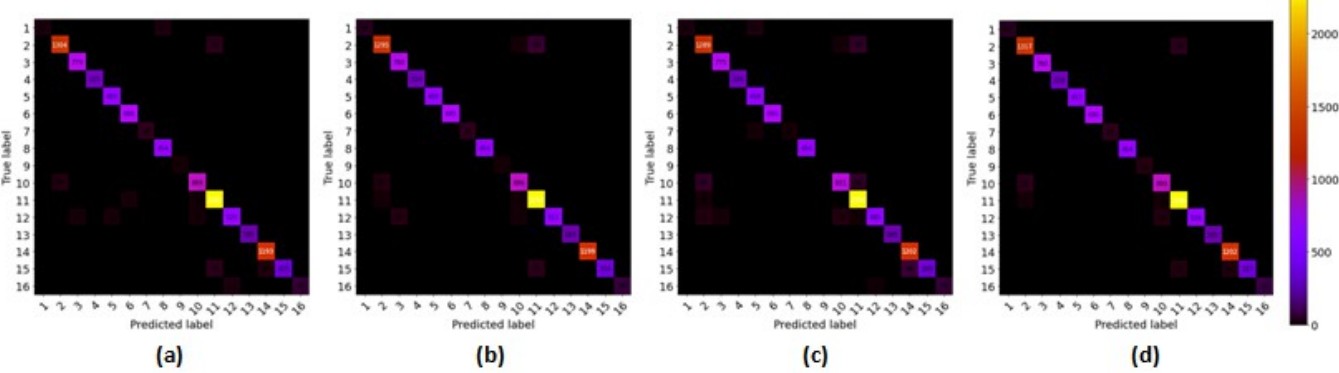

**Figure 13.** The confusion matrix of IP dataset: (**a**) R-HybridSN; (**b**) HybridSN; (**c**) SSRN; (**d**) HybridGBN-SR.

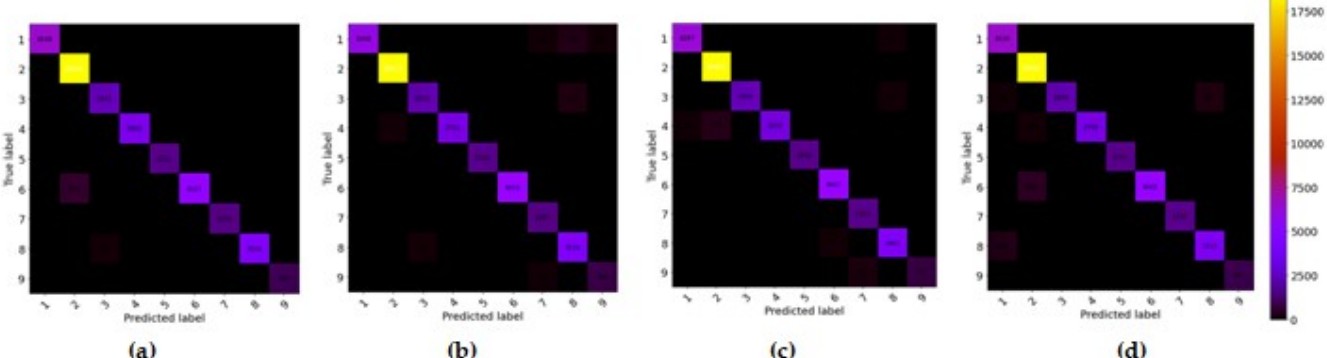

**Figure 14.** The confusion matrix of UP dataset: (**a**) R-HybridSN; (**b**) HybridSN; (**c**) SSRN; (**d**) HybridGBN-SR.

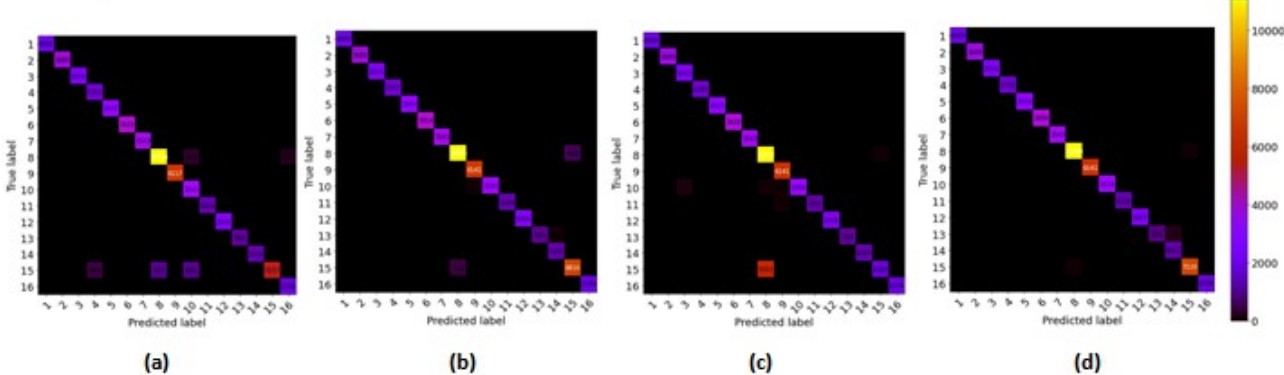

**Figure 15.** The confusion matrix of SA dataset: (**a**) R-HybridSN; (**b**) HybridSN; (**c**) SSRN; (**d**) HybridGBN-SR.

### 3.4.5. Classification Diagrams

We observe in Figures 16–18 that the SSRN, HybridSN, and R-HybridSN have more noisy scattered points in the classification maps, unlike the proposed HybridGBN-SR method over the IP, UP, and SA datasets. Therefore, the proposed method can remove the noisy scattered points and leads to smoother classification results without blurring the boundaries than the compared models when subjected to less training sample data.

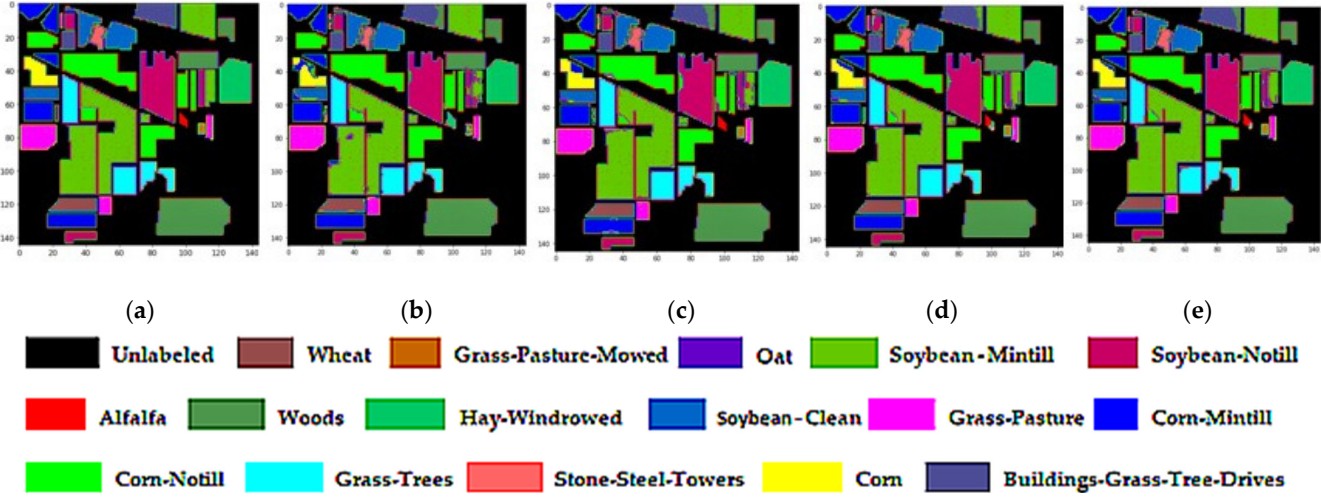

**Figure 16.** Classification maps of IP dataset: (**a**) Ground truth; (**b**) R-HybridSN; (**c**) HybridSN; (**d**) SSRN (**e**) HybridGBN-SR.

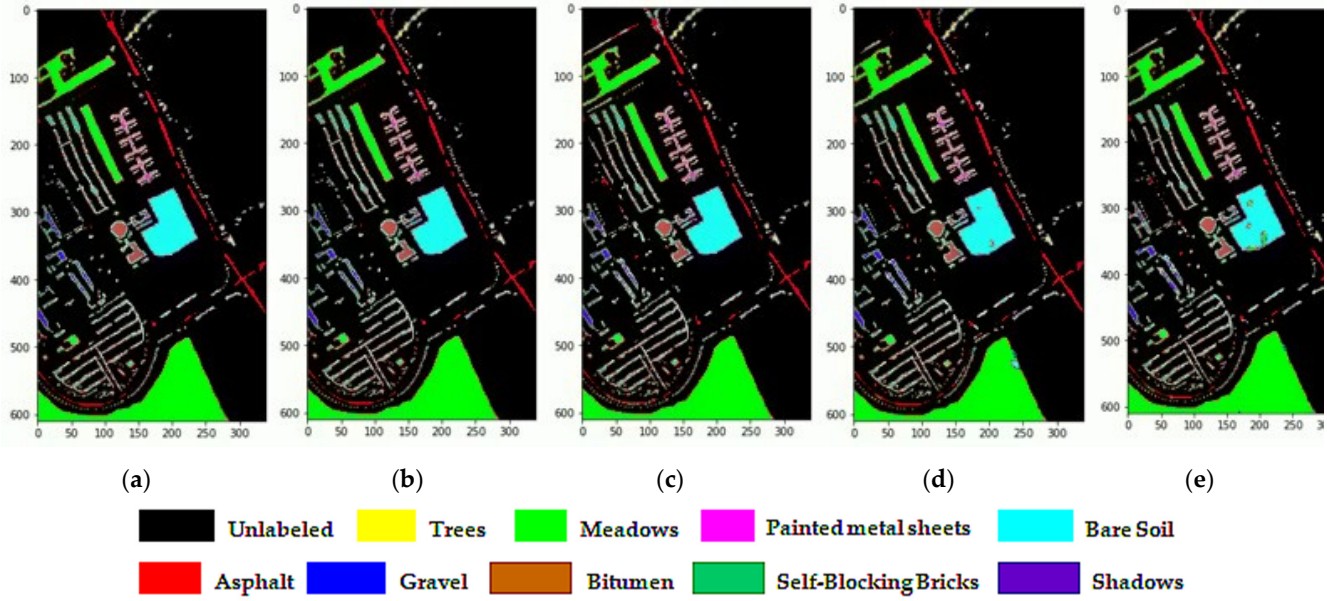

**Figure 17.** Classification maps of UP dataset: (**a**) Ground truth; (**b**) R-HybridSN; (**c**) HybridSN; (**d**) SSRN (**e**) HybridGBN-SR.

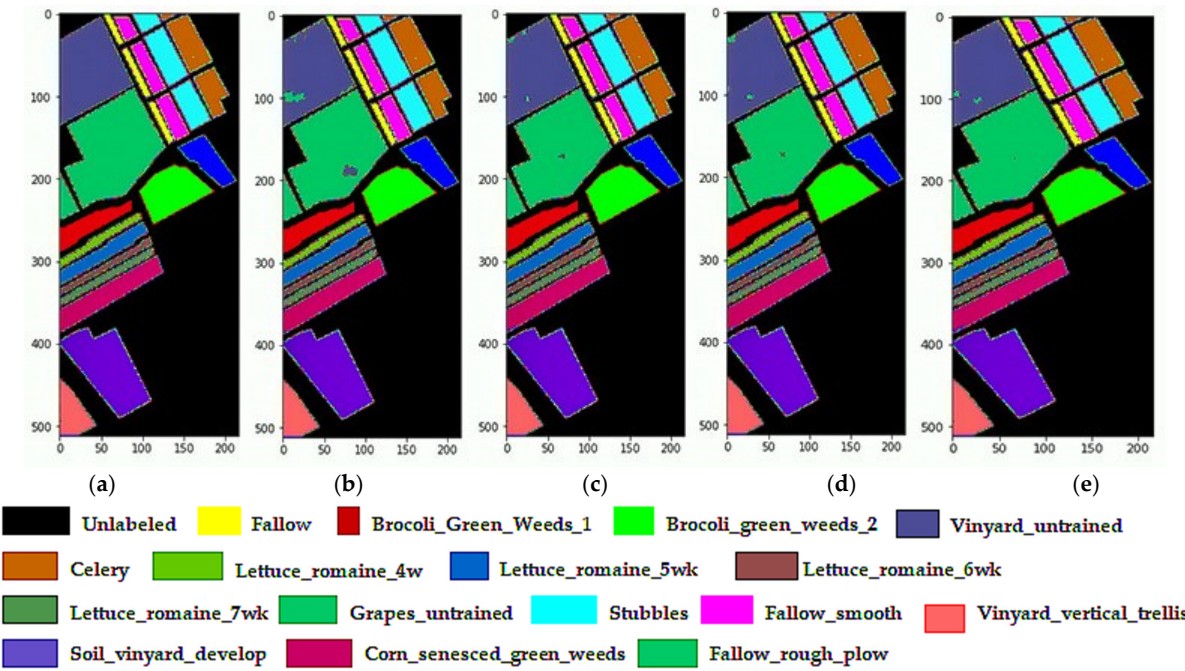

**Figure 18.** Classification maps of SA dataset: (**a**) Ground truth; (**b**) R-HybridSN; (**c**) HybridSN; (**d**) SSRN (**e**) HybridGBN-SR.

### 3.5. Varying Training Sample Data

To further compare the performance of the proposed Hybrid-SR with the selected state-of-the-art models, we varied the training sample data. We randomly trained the models at 2%, 5%, 10%, and 20% of the IP dataset, and 0.4%, 0.8%, 1%, 2%, and 5% of the UP and SA datasets, and then tested the models on the remaining data portion. The purpose of these experiments was to observe the variation trend and sensitivity of OA with the changing amount of training samples of the proposed HybridGBN-SR model compared with the selected state-of-the-art methods over the IP, UP, and SA datasets. The result is summarized in Tables 7–9.

**Table 7.** The effect of varying the training sample data for the SSRN, Hybrid, R-HybridSN, and HybridGBN-SR models on the overall accuracy (OA) over the IP dataset.

| Model | \multicolumn{5}{c}{Training Sample Data in Percentage} |
|---|---|---|---|---|---|
|  | 20% | 10% | 8% | 5% | 2% |
| 2D-CNN | 91.23 ± 0.21 | 83.86 ± 1 | 82.43 ± 0.62 | 75.47 ± 0.81 | 67.13 ± 1.12 |
| M3D-DCNN | 90.03 ± 2.18 | 80.1 ± 4.56 | 78.04 ± 2.13 | 68.88 ± 3.77 | 62.28 ± 3.18 |
| HybridSN | 99.3 ± 0.18 | 97.66 ± 0.23 | 96.37 ± 1.19 | 94.24 ± 1.01 | 83.14 ± 1.6 |
| R-HybridSN | 99.52 ± 0.16 | 98.44 ± 0.44 | 98.12 ± 0.35 | 96.46 ± 0.33 | 86.67 ± 1.02 |
| SSRN | 98.91 ± 0.12 | 97.25 ± 0.35 | 96.33 ± 0.41 | 93.39 ± 0.43 | 84.3 ± 1.61 |
| HybridGBN-SR | 99.3 ± 0.2 | 98.62 ± 0.22 | 98.31 ± 0.26 | 97.42 ± 0.22 | 91.44 ± 0.39 |

**Table 8.** The effect of varying the training sample data for the SSRN, Hybrid, R-HybridSN, and HybridGBN-SR models on the overall accuracy (OA) over the UP dataset.

| Model | \multicolumn{5}{c}{Training Sample Data} |
|---|---|---|---|---|---|
|  | 5% | 2% | 1% | 0.80% | 0.40% |
| 2D-CNN | 96.59 ± 0.21 | 94.5 ± 0.4 | 91.82 ± 0.56 | 89.98 ± 0.38 | 85.27 ± 0.90 |
| M3D-DCNN | 92.8 ± 0.95 | 89.27 ± 1.35 | 87.19 ± 1.71 | 82.75 ± 2.84 | 76.53 ± 3.94 |
| HybridSN | 99.45 ± 0.09 | 97.86 ± 0.56 | 95.86 ± 0.93 | 93.3 ± 1.41 | 85.95 ± 1.58 |
| SSRN | 99.57 ± 0.13 | 99.07 ± 0.17 | 97.67 ± 0.4 | 97.12 ± 0.28 | 93.41 ± 0.77 |
| R-HybridSN | 99.47 ± 0.14 | 98.47 ± 0.27 | 96.4 ± 1.66 | 95.64 ± 0.52 | 91.60 ± 1.12 |
| HybridGBN-SR | 99.54 ± 0.07 | 99.13 ± 0.17 | 97.85 ± 0.4 | 97.33 ± 0.45 | 94.14 ± 0.61 |

**Table 9.** The effect of varying the training sample data for the SSRN, Hybrid, R-HybridSN, and HybridGBN-SR models on the overall accuracy (OA) over the SA dataset.

| Model | Training Sample Data | | | | |
|---|---|---|---|---|---|
| | 5% | 2% | 1.00% | 0.80% | 0.40% |
| 2D-CNN | 96.63 ± 0.24 | 94.67 ± 0.15 | 93.55 ± 0.26 | 93.03 ± 0.26 | 91.38 ± 0.44 |
| M3D-DCNN | 92.65 ± 0.49 | 90.17 ± 0.56 | 88.02 ± 1.35 | 86.82 ± 1.18 | 83.42 ± 1.6 |
| HybridSN | 99.83 ± 0.1 | 99.57 ± 0.25 | 98.72 ± 0.59 | 97.78 ± 0.78 | 94.88 ± 0.9 |
| R-HybridSN | 99.82 ± 0.04 | 99.36 ± 0.14 | 98.25 ± 0.4 | 96.97 ± 0.57 | 94.33 ± 0.48 |
| SSRN | 98.7 ± 0.51 | 98.02 ± 0.16 | 96.94 ± 0.55 | 96.87 ± 0.29 | 93.64 ± 0.22 |
| HybridGBN-SR | 99.94 ± 0.02 | 99.72 ± 0.11 | 99.34 ± 0.14 | 98.37 ± 0.43 | 95.8 ± 1.19 |

We observe in Tables 7–9 that the proposed HybridGBN-SR model has better overall accuracy than the state-of-the-art models in almost all the training sample data splits. Figure 19 illustrates that as the training sample data are reduced, the classification accuracy gap between the proposed HybridGBN-SR model and the selected state-of-the-art models widens, demonstrating different reduction speeds among the compared models. For instance, in the IP dataset (see Table 7 and Figure 19a), at 8% of the training sample data, the HybridGBN-SR improves the overall accuracy of 2D-CNN, M3D-CNN, SSRN, HybridSN, and R-HybridSN by +15.88%, +20.27%, +1.98%, +1.94%, and +0.19%, respectively. In comparison, at 2% training sample data, the HybridGBN-SR improves the overall accuracy of 2D-CNN, M3D-CNN, SSRN, HybridSN, and R-HybridSN by +24.31%, +29.16%, +7.14%, +8.3%, and +4.77%, respectively.

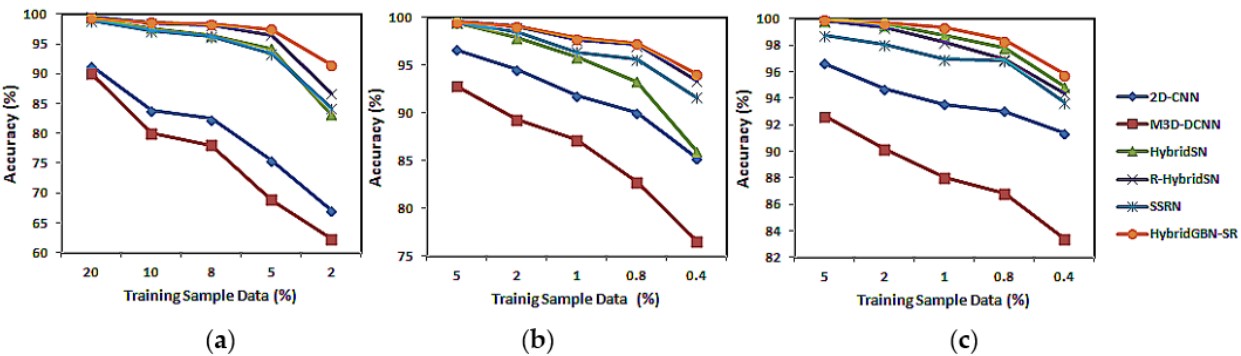

**Figure 19.** Varying training sample data for (**a**) IP; (**b**) UP; (**c**) SA datasets.

In the SA dataset (See Table 9 and Figure 19c), at 5% and 0.04% training sample data, the proposed HybridGBN-SR model increases the overall accuracy (OA) of the second-best model (HybridSN) by +0.11% and +0.92, respectively. It shows an increase in the performance gap between our model and other models as the training sample data drastically reduce. The same trend is observed in the UP dataset (See Table 8 and Figure 19b). Therefore, we can conclude that the robustness of the proposed HybridGBN-SR model is more pronounced as the amount of training portion decreases across all the experimental datasets. This implies that the proposed HybridGBN-SR model can extract sufficient discriminative features even at minimal training sample data. We attribute this to the genomic residue connection in the design of the HybridGBN-SR model.

*3.6. The Time Complexity of the Selected Models over IP, UP, and SA Datasets*

Table 10 summarizes the training and testing time in seconds of SSRN, HybridSN, R-HybridSN, and the proposed HybridGBN over the IP dataset on 5% training and 95% testing sample data, and over UP and SA datasets on 1% training and 99% testing sample data.

The computational efficiency in training and testing time (in seconds) shown in Table 10 indicates that the proposed HybridGBN-SR model performs better than SSRN, is comparable with R-HybridSN, and worse than HybridSN over the three datasets. However, we observe that the training and testing time of the deep learning model is related to the

experimental environment, model structure, number of training epochs, amount of training samples, patch size, etc. For instance, the HybridSN model trains and tests faster than the other models due to its simple network structure. The SSRN is the slowest in training and testing because it contains a deep network structure and takes a long time to learn (number of epochs). Lastly, the speed difference between the HybridGBN-SR and R-HybridSN can be attributed to the network optimization parameters.

**Table 10.** The training and testing time in seconds over IP, UP, and SA datasets using SSRN, HybridSN, R-HybridSN, and HybridGBN-SR.

| Dataset | SSRN | | HybridSN | | R-HybridSN | | HybridGBN-SR | |
|---|---|---|---|---|---|---|---|---|
| | Train | Test | Train | Test | Train | Test | Train | Test |
| IP | 91.1 | 2.6 | 31.9 | 3.2 | 23.1 | 2.6 | 43.6 | 3.4 |
| UP | 108.9 | 7.3 | 12.4 | 6.9 | 30.1 | 9.4 | 21.3 | 6.2 |
| SA | 122.6 | 12.3 | 13.2 | 8.9 | 16.4 | 12.3 | 30.2 | 12.9 |

## 4. Conclusions

This work seeks to further the scientific work of developing deep networks for HSI classification. We propose a deep 3D/2D genome graph-based network (abbreviated as HybridGBN-SR) model that extracts discriminative spectral–spatial features from very few training samples. In the network design, the proposed HybridGBN-SR model uses the Genoblocks, a concept borrowed from biological graph genomes. The Genoblocks innovatively utilize multi-scale kernels, identical and non-identical residual connections, to extract abundant contextual features vital to attaining a high generalizability rate. The residual connections promote the backpropagation of gradients to extract more discriminative features and prevent overfitting, leading to high classification accuracy. The proposed HybridGBN-SR model achieves reduced computational cost by replacing the Geno3Dblock with the low-cost Geno2Dblock at the top of the network structure. The Geno2Dblock contains dilated 2D convolutions to extract further discriminative HSI features, resulting in increased model computational efficiency while maintaining classification accuracy. The proposed HybridGBN-SR model's robustness is evidenced by its better convergence than SSRN and HybridSN across all the datasets and its ability to achieve better classification accuracy with a small number of training samples as compared to the state-of-the-art methods such as SSRN, HybridSN, and R-HybridSN over the IP, UP, and SA.

**Author Contributions:** Conceptualization, H.C.T., E.C., R.M.M. and D.O.N.; software, H.C.T. and D.O.N.; resources, E.C.; writing—original draft preparation, H.C.T., D.O.N. and R.M.M.; writing—review and editing, H.C.T., E.C., L.M., D.O.N. and R.M.M.; supervision, E.C. and L.M.; funding acquisition, E.C. All authors have read and agreed to the published version of the manuscript.

**Funding:** This work was supported in part by the National Natural Science Foundation of China under Grants U1804152 and 62101503.

**Institutional Review Board Statement:** Not applicable.

**Informed Consent Statement:** Not applicable.

**Data Availability Statement:** All datasets used in this research are openly accessible online (http://www.ehu.eus/ccwintco/index.php?title=Hyperspectral_Remote_Sensing_Scenes (accessed on 15 February 2022)).

**Acknowledgments:** The authors express gratitude to http://www.ehu.eus (accessed on 15 February 2022) for publicly providing the original hyperspectral images to advance research remote sensing.

**Conflicts of Interest:** The authors declare no conflict of interest.

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
