# Peer review of "HybridGBN-SR: A Deep 3D/2D Genome Graph-Based Network for Hyperspectral Image Classification"

_remotesensing, doi:10.3390/rs14061332_

Round 1
Reviewer 1 Report
I I wish to commend the authors for good paper presentation. However, I have minor edit
- Line 78: add (CNN) asin convolutional neural network (CNN)
- line 80: The aim of the CNN is {delete "main" and convolutional neural network}
- all "et al" should be in italics "et al"
- Line 228-229: use CNN. no need repeating convolutional neural network
- reference section should be in MDPI format and template
Author Response
Dear Reviewer,
Thank you for allowing a resubmission of our manuscript, with an opportunity to address the comments you raised. Kindly find our point-by-point response to the comments in the attached file.
Kind regards,

Reviewer 2 Report
Overall, this manuscript is well-written and easy to understand. The carried out literature review is relatively decent. However, many of the supplied references are not that recent. The proposed methodology is clear and supplemented with enough codes, equations and figures to aid the reader in comprehension. The authors compare their proposed algorithm with the state-of-the-art and their results good. Finally, their conclusions are supported by their results and reflect good performance.
I only have the following comments:
- Missing spaces in lines 41, 73, 74, 76, 221, 250 and 362.
- Almost all the figures need to be provided at a higher resolution.
Author Response

(The authors gave the same response as above.)

Reviewer 3 Report
Globally, the manuscript is very well written and organized. However, there are some minor typos and editing errors that must be corrected before the final publication of the manuscript. Please refer to the attached commented PDF document where these bugs are highlighted.
As a general comment, that spread over the whole document, authors should define every acronym/abbreviation the first time they use it (e.g., HSI, HSIC, etc.), and once defined use it without redefining it. Additionally, authors should always use capital (or lower) first characters of the terms/concepts they define and use in the manuscript (e.g., genoblocks/Genoblocks, hybridGBN…/HybridGBN…, etc.).
All figures (and tables) must be explicitly referenced in the main text (e.g., figure 1, figure 2, etc.).
Extra care should be payed to the mathematical equations/expressions! Do not repeat the use of the same char (e.g., “b”) to mean different things. When using single/isolated mathematical chars in the main text (between words) write them as math symbols. Some examples are also highlighted in the attached PDF document.
Lines 326-327: “In this research, ?? (.) is a MaxPooling2D of size 3.”. Why “size 3”? If there isn’t another value that could be used, you don’t need to justify the size…

Author Response
Dear Reviewer,
Thank you for allowing a resubmission of our manuscript, with an opportunity to address the comments you raised. Kindly find our point-by-point response to the comments.
Kind regards,

Reviewer 4 Report
Language should be checked: while grammatically correct, some sentences are questionable. E.g.: L18 "to train less training data": the algorithm is trained, not the data. The sentence should read "to use a smaller training dataset". L59 "The resulting raw HSI images contain a wide amount of data": a large amount of data? The introduction is too long and the initial part, L31 to L77, unnecessary. L19 acronyms should be explained at first use: HSI. L52 "electronic spectrum" -> "electromagnetic spectrum" L57 "Since the HSI system uses both imaging and spectroscopic methods to spatially locate specific components within the image scene under investigation based on their spectral features.": is part of the sentence missing? L61 "voluminous spectral": information? L65 What is HSIC? L68 Chen et al. [12] uses a 7x7 neighbor region, not 27x27. L90 "acrylic"? L105-118 is in part a repetition of L78-104, with more technical details. I suggest to merge the two blocks. L171 What is "w"? The number of spectral components in I? L191 "use first w columns of D while restricting E to first w eigenvalues" after re-ordering them. L282 (lowercase) w, k, p and s are not in eq. 5: do they refer to the corresponding uppercase letters? What is "F" in eq. 5? L298 Figure 7 seems to be identical to figure 6, apart for the label at the top. I refers to a modified third connection with respect to the one in Fig. 5: is there a way to differentiate the connection graphically? Otherwise I think Fig. 7 is useless. L331 acronyms should be explained at first use: FC. L336 Eq. 9: describe the parameters. L361 "The ground truth of the Indian Pines scene dataset consists of 16 not mutually exclusive[37].": classes? Labels? L545 - L548 The two sentences referring to the IP and UP+SA datasets seems to be identical: why have the operations on the datasets reported in two different sentences?Author Response
Dear Reviewer,
Thank you for allowing a resubmission of our manuscript, with an opportunity to address the comments you raised. Kindly find our point-by-point response to the comments.
Kind regards,
